



# Impacts from cascading multi-hazards using hypergraphs: a case study from the 2015 Gorkha earthquake in Nepal

Alexandre Dunant[1*], Tom R. Robinson[2], Alexander L. Densmore[1], Nick J. Rosser[1], Ragindra Man Rajbhandari[3], Mark Kincey[4], Sihan Li[5], Prem Raj Awasthi[3], Max Van Wyk de Vries[6,7], Ramesh Guragain[8], Erin Harvey[1] and Simon Dadson[9]

[1] Institute of Hazard, Risk, and Resilience and Department of Geography, Durham University, Durham, UK

[2] School of Earth and Environment, University of Canterbury, Christchurch, New Zealand

[3] UN Resident Coordinator's Office, Nepal

[4] School of Geography, Politics, and Sociology, Newcastle University, Newcastle, UK

[5] Department of Geography, University of Sheffield, Sheffield, UK

[6] Department of Geography, University of Cambridge, Cambridge CB2 3EL, UK

[7] Department of Earth Sciences, University of Cambridge, Cambridge CB3 0EZ, UK

[8] National Society for Earthquake Technology-Nepal (NSET), Nepal

[9] School of Geography and the Environment, University of Oxford, UK

[*] Corresponding author: alexandre.dunant@durham.ac.uk

## Abstract

This study introduces a new approach to multi-hazard risk assessment, leveraging hypergraph theory to model the interconnected risks posed by cascading natural hazards. Traditional single-hazard risk models fail to account for the complex interrelationships and compounding effects of multiple simultaneous or sequential hazards. By conceptualising risks within a hypergraph framework, our model overcomes these limitations, enabling efficient simulation of multi-hazard interactions and their impacts on infrastructure. We apply this model to the 2015 $M_w$ 7.8 Gorkha earthquake in Nepal as a case study, demonstrating its ability to simulate the primary and secondary effects of the earthquake on buildings and roads across the whole earthquake-affected area. The model predicts the overall pattern of earthquake-induced building damage and landslide impacts, albeit with a tendency towards over-prediction. Our findings underscore the potential of the hypergraph approach for multi-hazard risk assessment, offering advances in rapid computation and scenario exploration for cascading geo-hazards. This approach could provide valuable insights for disaster risk reduction and humanitarian contingency planning, where anticipation of large-scale trends is often more important than prediction of detailed impacts.

## Keywords

Cascading multi-hazards, multi-hazard modelling, earthquake impacts, landslides, Nepal, network modelling, hypergraphs



## 1. Introduction

There is a growing recognition over the last 15 years that natural hazards can interact and occur in conjunction with each other, leading to a potential compounding effect that is greater than the sum of the single-hazard impacts (Kappes et al., 2012; Terzi et al., 2019). While the global prevalence of cascading hazards specifically is difficult to quantify reliably, there are increasing calls for effective multi-hazard risk assessments (e.g., Ward et al., 2022). Multi-hazards are defined by UNISDR (2016) as "events [that] may occur simultaneously, cascadingly or cumulatively over time, and taking into account the potential interrelated effects". Multi-hazard approaches seek to overcome the limitations of a narrower focus on single-hazard models, which are unable to account for the observed inter-relationships between different hazards as well as potential compounding or cascading effects (e.g., Gill and Malamud, 2014; Tilloy et al., 2019; Dunant, 2021; Ming et al., 2022). Multi-hazard approaches to risk are now widely encouraged (e.g., UNISDR, 2005; Government Office for Science, 2012) and are increasingly integrated into risk assessment (see recent reviews by Gill et al., 2022; Ward et al., 2022).

There remain, however, some important challenges and limitations with multi-hazard risk assessment. Because of the difficulties in recognising, understanding, and defining the inter-relationships between hazards, and the lack of data on their co-dependence (Tilloy et al., 2019; Hochrainer-Stigler et al., 2023), most 'multi-hazard risk' models simply overlay single hazards without considering their interactions – an approach that Gill and Malamud (2014) termed 'multi-layer single hazard'. Even when hazard-hazard interactions are considered in risk models, there is still a lack of comprehensive approaches that capture the intricate interplay among hazards, exposure, and vulnerability beyond simple spatial overlaps (Mignan et al., 2014; de Ruiter et al., 2020). These interactions are critical because of the possibility that risks may be clustered in space and time or may amplify each other, as demonstrated by Mignan et al. (2014). Zschau (2017) extended the ideas of Gill and Malamud (2014) to risk assessment, distinguishing between risk from single hazards, risk from multi-layer single hazards, and risk from multi-hazards – the latter allowing for dynamic hazard interactions, but no dynamic interactions between hazard and exposure or vulnerability). Hochrainer-Stigler et al. (2023) noted that hazard-exposure relationships and changes in exposure over time, as well as vulnerability, are also critical to fully characterise multi-risks. This complexity means that multi-hazard risk modelling can be both computationally expensive and extremely demanding of quality input data (e.g., Kappes et al. 2012). Multi-hazard risk models may also be limited by the diversity of hazard types that can be incorporated, mismatches in the appropriate spatial and temporal scale of analyses, and complex data requirements (e.g., Kappes et al., 2012; Tilloy et al., 2019; Dunant, 2021).

A further complication is the growing need for national, regional, or even global-scale risk assessments, in order to understand potential patterns of impacts, provide science-based evidence for disaster risk reduction and advocacy, and allow coordinated planning (see review by Ward et al., 2020). At the same time, data are available at ever-increasing spatial and temporal resolution, including information on populations, building stock, and topography, as well as datasets on hazard drivers such as rainfall forecasts or observed precipitation. While these are welcome developments, the combination of demands for increasing scale and increasingly-fine spatial and temporal resolution data leads to a much higher computational burden. Addressing the need for both larger spatial scales and finer spatio-temporal resolutions is a growing challenge for the assessment of multi-hazard risks. The distribution of risk may also be highly spatially imbalanced if exposed elements are concentrated in specific areas, meaning that grid-based or GIS-based approaches to risk modelling may expend much computational effort on areas where risk is low or negligible.



To address these concerns, Dunant et al. (2021a) proposed a novel approach to multi-hazard risk modelling using graph
theory. In this framework, both the hazards and the elements at risk are modelled as a set of interconnections between
nodes. For example, a house can be linked to ground accelerations in an earthquake, or a hillslope to rainfall in a storm.
This framework can then be used to generate many disaster scenarios by cascading from node to node according to a set
of rules (e.g., a threshold earthquake shaking value for slope failure). The resulting network model is highly
computationally efficient, and the network structure is a natural fit to the simulation of coincident or cascading events
and their propagation through exposure networks (Dunant et al., 2021a) because network structures are purposefully
designed to capture the interdependencies and feedbacks among different elements. The framework is agnostic to the
types of objects that can be included, so it can be easily adapted to include hazard-hazard, hazard-exposure, and
hazard-vulnerability relationships. It is also highly flexible, so that the links between objects can be represented via
different interactions depending on the level of understanding and data availability, including threshold values,
empirical functions, fuzzy distributions, process models, or other approaches (e.g., Tilloy et al., 2019).
Despite its advantages, however, the network model suffers from some important limitations. Most critically, because
the interactions in a network model are modelled as pairs, the computational burden grows substantially as the number
of components (nodes and edges) of the model increases. Prior applications focused on the epicentral area of the 2016
$M_w$ 7.8 Kaikōura earthquake (Dunant et al., 2021a) and the area around Franz Josef township (Dunant et al., 2021b),
both in New Zealand and containing on the order of hundreds of nodes. Expanding the network model to a national
scale at a similar resolution would increase the model size by several orders of magnitude. Similarly, increasing the
number of hazards that are considered would lead to a combinatorial increase in interactions and rapid growth in
computation time.
Here we propose a new approach to modelling the impacts of multi-hazards using hypergraphs – two-dimensional
surface equivalents of the pairwise links found in the graph-theory network model of Dunant et al. (2021a). The
hypergraph model retains the advantages of the network approach while simultaneously reducing the model complexity.
Below, we first present a brief review of graphs and hypergraphs and outline the benefits of using hypergraphs in a
multi-hazard risk modelling framework. We describe the structure of the multi-hazard impact model, including its
components and the interactions between nodes. We illustrate its application by simulating the impacts from the 2015
$M_w$ 7.8 Gorkha earthquake in Nepal, as an exemplar of a large-scale event that had cascading effects on people and
infrastructure due to both primary and secondary hazards. We close by considering wider potential applications of the
hypergraph model, including national- or regional-scale disaster scenario ensembles and how they might be used to
support humanitarian contingency planning (e.g., Robinson et al., 2018).

### 2. Summary of graph and hypergraph approaches

A graph is essentially a mathematical representation of a network. The term was originally introduced by Sylvester
(1878) but graph theory had been used more than a hundred years before by Euler (1736) to solve the Seven Bridges of
Königsberg problem. Since then, graph theory has been used in a wide variety of fields such as geography, computer
science, social science, and biology (e.g., Buzna et al., 2006; Chorley & Kennedy, 1971; Dezső & Barabási, 2002;
Dorogovtsev & Mendes, 2003).





A graph comprises a set of nodes connected by edges. In the context of risks posed by environmental hazards, such
nodes may represent a geographical location (spatially explicit; e.g., a fault segment, or a house) or a nominal property
(spatially implicit; e.g., the occurrence of an earthquake) and the edges represent the relations between the nodes (e.g.,
earthquake shaking affecting exposed houses) (Fig. 1A).

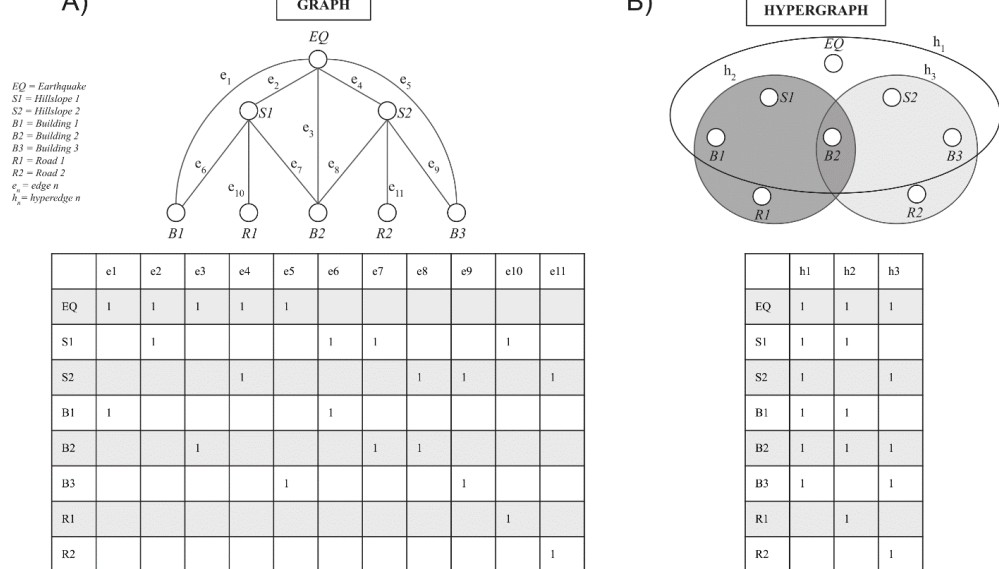

**Figure 1: Graph (A) and hypergraph (B) representations of a hypothetical set of hazard and exposure interactions. The same**
**set of elements are represented in both graphical form (top) and tabular form as incidence matrices (bottom). In the tables, a**
**blank cell means no interaction between the nodes, and a value of 1 means that interactions are possible between the nodes.**

A defining characteristic of graphs is the set of pairwise connections or edges between nodes that define the
relationships between these nodes. For example, we would represent earthquake shaking on a set of hillslopes as edges
between the earthquake and each hillslope that is affected. In tabular form, each edge is represented by a row in a
relational database, called an incidence matrix (Fig. 1A). The edges are directional, so a two-way relationship – for
example, a hillslope potentially affecting a road via landslides, and a road potentially affecting a hillslope via excavation
and steepening – would be represented by two separate rows.

As summarised by Dunant et al. (2021a), here we consider relationships between nodes that are observed or felt – that
is, via shaking, mass movement, or water flow. We also consider that nodes are connected if (1) the geographical effect
of one node overlaps that of another, and (2) that effect is relevant to considering impacts from hazards. For example,
earthquake ground shaking might affect a hillslope and trigger a new landslide or the mobilisation of loose material in a
debris flow; to allow for these effects, we would represent the relationship between earthquake and the hillslope as an
edge, and the relationship between the hillslope and any houses or road segments on it as a series of additional edges
(Fig. 1A). If we were to assume that the earthquake ground motion can potentially cause direct impacts on houses but
not roads, then the earthquake would be connected to the houses by edges but not to the road segments (Fig. 1A).



In contrast, a hypergraph is a special type of graph where the edges, called hyperedges, can link one or more nodes (Fig.
1B). This allows us to represent interactions that extend beyond a single pair of nodes (Wolf et al., 2016). Compared to
pairwise edges, which only connect two nodes, hyperedges can connect multiple nodes and provide a more natural
representation for the spatial overlap between exposed elements, like houses, and geographical hazard footprints.
Hyperedges can thus represent nested information between the nodes of the system, such as their properties or locations,
with far fewer tabular entries (Fig. 1B). The hypergraph uses fewer edges to represent the same number of interactions
for a given number of nodes; this size difference (e.g., for the example in Figure 1, 11x8=88 entries for the graph
framework and 3x8=24 for the hypergraph framework) highlights the efficiency of the hypergraph approach.
The increased efficiency enabled by hypergraphs becomes more apparent when dealing with large, interconnected
datasets and when iterative data manipulation is required. For example, we can run hundreds or thousands of separate
simulations on the same hypergraph, choosing different events or altering input parameters within a Monte Carlo
framework (e.g., Dunant et al., 2021a) to generate ensemble distributions of scenario outcomes (Robinson et al., 2018).
The improvement in computation time allows the hypergraph framework to be applied to multi-hazards risk assessment
over larger extents, over longer time periods, and with more complex interactions than would be feasible using a
GIS-based approach or standard graph framework.
**3. Methodology**
Below we describe the setup and operation of the multi-hazard hypergraph model and describe its application to the
2015 Gorkha earthquake.
**3.1 Model overview and setup**
The model is based around a set of interactions between elements in Nepal that are drawn from experience in both the
annual monsoon (Kincey et al., 2022; Jimee et al., 2019; Goda et al., 2015; Rosser et al., 2021; Kargel et al., 2016) and
recent earthquakes, including the 2015 Gorkha event (e.g., Roback et al., 2018; Milledge et al., 2019; Kincey et al.,
2021). For the simulations in this paper, the model is driven only by earthquakes (Fig. 2) and seeks to assess the risk to
buildings and roads at a national scale. Earthquake shaking is simulated as a spatial distribution of peak ground
acceleration (PGA) values; these could be derived from measurements or generated for a potential scenario earthquake
via a shaking model. For the experiments shown here, we use empirical PGA values estimated by the US Geological
Survey Shakemap for the 2015 Gorkha earthquake
(https://earthquake.usgs.gov/earthquakes/eventpage/us20002926/shakemap/pga). Earthquake shaking can affect
infrastructure either directly (described via a set of fragility functions) or by triggering landslides. Landslides, in turn,
may affect both buildings and roads. In this version of the model, other hazards such as rainfall and floods are not
considered, but they could be added via additional sets of hyperedges and interactions.



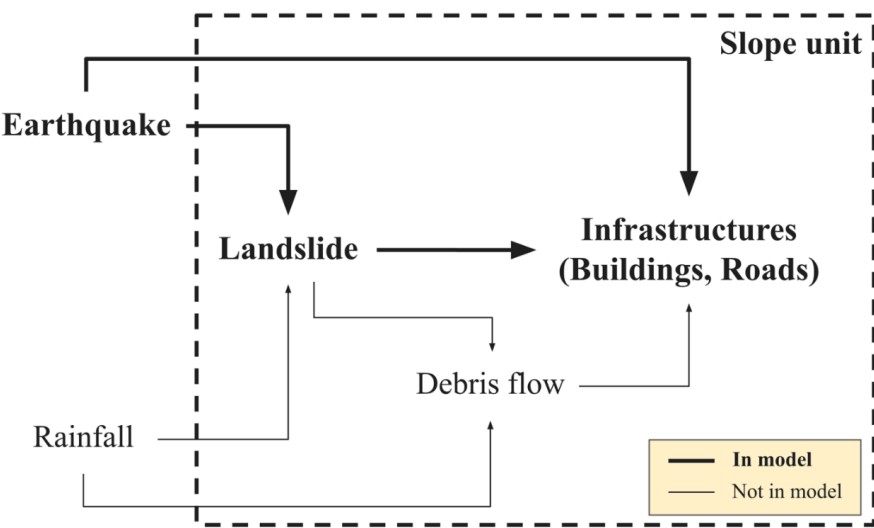


**Figure 2: Driving stimuli and important process interactions for the area affected by the 2015 Gorkha earthquake in Nepal.**
**The elements that are included in the multi-hazard impact experiments documented here are shown in bold text.**


To model coseismic landslides, we subdivide the landscape into discrete units and consider the characteristics of the
topography as well as the driving mechanisms within those subdivisions. Here we divide the landscape into slope units
that are bounded by drainages and divide lines (Alvioli et al., 2016; Woodard et al., 2024) (see Supplemental
Information and Fig. S1). Woodard et al. (2024) demonstrated that slope units are preferable to gridded topography
when representing landslide susceptibility, especially for input landslide data that are imprecise or highly spatially
variable in quality.

The hyperedges are constructed based on the interactions in Figure 2. A hyperedge connects the earthquake node with
all of the slope units and buildings within the 'footprint' of the earthquake, defined by the extent of a minimum PGA (X
g) contour. Similarly, hyperedges connect each slope unit with the buildings and roads (divided into 100 m segments)
within it; we therefore assume that landslides from one slope unit cannot impact elements in another. Attributes for each
building, road segment, and slope unit, such as location, PGA, building type, landslide susceptibility, are stored on the
hyperedges and can be displayed as continuous values in a tabular form. We describe each of these attributes below.

We use building locations and roads taken from the Humanitarian OpenStreetMap Team, covering the whole of Nepal,
and available at https://data.humdata.org/dataset/hotosm_npl_buildings and
https://data.humdata.org/dataset/hotosm_npl_roads, respectively (accessed 1 January 2021). The datasets contain c. 7.1
million building polygons and c. 3 million road segments. Because we lack specific information on the construction
type of each building to assess its fragility, we instead use exposure data from the Modeling Exposure Through Earth
Observation Routines (METEOR) project (https://maps.meteor-project.org/map/building-exposure-map-of-nepal)
(version 2020-02-15) , which includes a list of building types and the number and value of each type within each cell of
a 90 x 90 m grid across Nepal. The PGA value of the 2015 Gorkha earthquake is extracted at the centroid of each
METEOR grid cell. To account for variability in construction detail and quality within these broad types, we adopt low,



middle, and high fragility functions for the 'complete damage' state for typical building types in Nepal from the METEOR dataset (Fig. 3). We take the definition of 'complete damage' from the Hazus framework of the US Federal Emergency Management Agency (FEMA, 2020). We generate a weighted-average fragility function for the buildings within each 90 x 90 m grid cell based on the proportion of different building types; thus, in the absence of any national-scale building-specific information, all buildings within that cell are assumed to have the same average fragility. We assess the likelihood of 'complete damage' because this implies loss of usability or habitability, with consequences for displacement and disruption to life and livelihoods, and is typically used to estimate fatality and injury rates (FEMA, 2020).

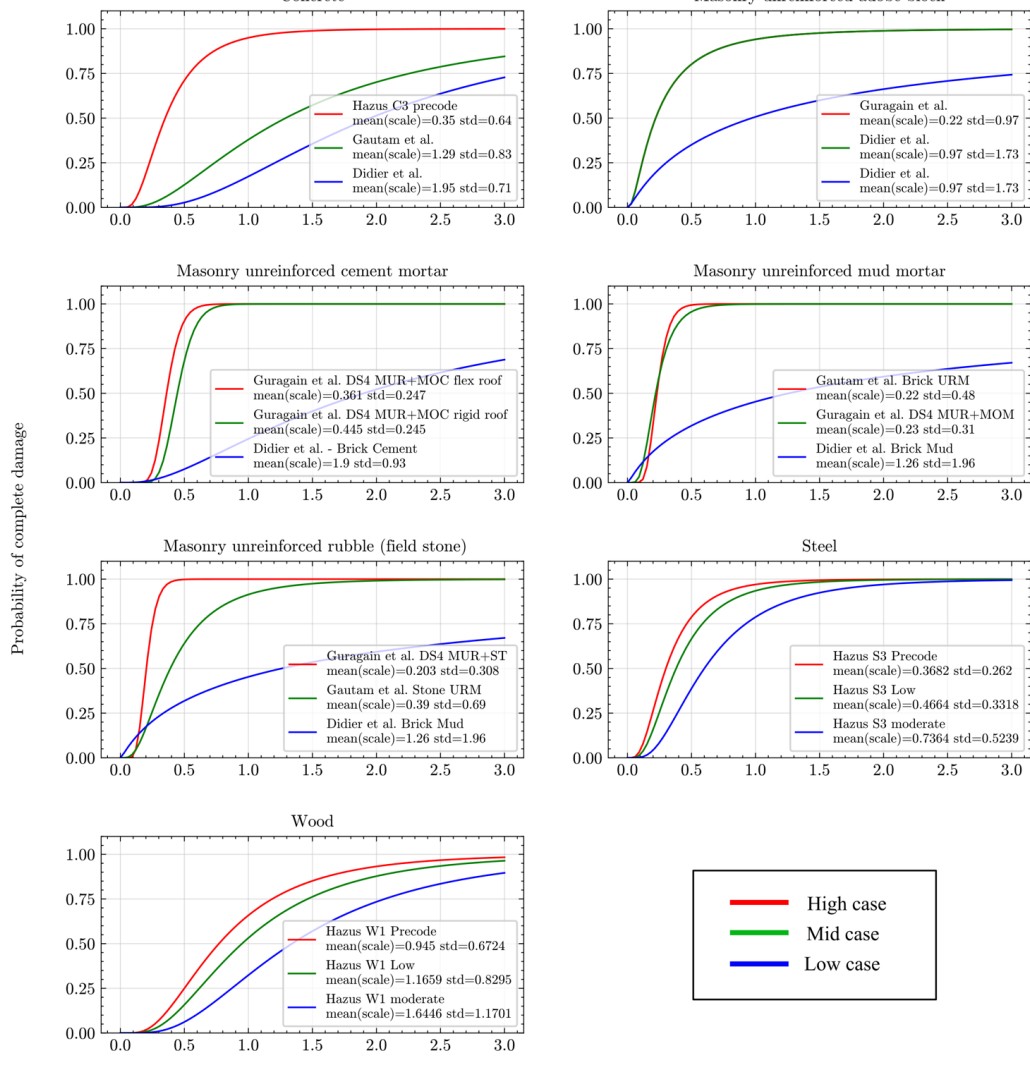





**Figure 3: Fragility functions used in the hypergraph network modelling. Each panel shows fragility curves for a different building type in the METEOR dataset, and which relate the peak ground acceleration (PGA, in g) to the probability of being reduced to a complete damage state. Note that each sigmoidal fragility curve is defined by two parameters: a mean or scale parameter that sets the PGA value for a 50% probability of complete damage, and a standard deviation (std) that defines the spread of the curve. Parameter values and sources for the fragility curves are included in the plots.**

We estimate landslide susceptibility based on topographic factors alone, using a seven-parameter static susceptibility model that incorporates elevation, hillslope aspect, distance to rivers, plan-view curvature, regional relief, local hillslope gradient, and a terrain ruggedness index. These factors are derived from a 10 m digital elevation model (DEM) that was downsampled from the 5 m Advanced Land Observing Satellite World 3D DEM (https://www.aw3d.jp/en/products/standard/). We generate the susceptibility model using a gradient boosting machine learning approach, XGBoost, implemented in Python. For the experiments shown here, the susceptibility model is trained on the locations of coseismic landslides triggered by the 2015 Gorkha earthquake as mapped by Kincey et al. (2021), yielding an area under the receiver operating characteristic (ROC) curve of 0.75 (Fig. S2). We stress that this susceptibility layer is used simply as an exemplar which is optimised for the 2015 Gorkha earthquake; for other model applications, susceptibility data generated with other approaches (see review in Reichenbach et al., 2018), or trained on different inventories, could be substituted. Because landslide susceptibility is modelled on a 10 x 10 m grid, each slope unit contains a unique distribution of cell-wise susceptibility values in the range [0,1], and each building polygon or road segment overlaps with one or more cellwise susceptibility values. Importantly, because the multi-hazard model is intended to simulate dynamic cascading scenarios, we choose not to include earthquake shaking as a determining factor in the static landslide susceptibility model. This choice preserves independence between shaking, landslide triggering, and the propagation of hazards along the hyperedges within the model.

We extract the mean and standard deviation of susceptibility for each slope unit, building and road segment, although other measures of the distribution could also be used. Because we lack general building or road fragility functions for landslides that are comparable to those for earthquakes and that encompass the wide range of possible landslide types and sizes (see Luo et al., 2023, for a recent review), we adopt a simplified binary vulnerability model, such that any building or road that is affected by a landslide is considered as 'impacted'.

**3.2 Simulation steps**

In each simulation, the model works iteratively through the hyperedges that connect the driving stimulus of earthquake shaking to the other elements in the model, checking against a condition to see whether that hyperedge of the network is 'activated' – i.e., a building is damaged by earthquake shaking, or a slope unit is affected by one or more landslides. Activation of that hyperedge then allows the stimulus to propagate, and potentially to cascade along other hyperedges if further conditions are met (Fig. 4). The simulation continues until all cascades stop and no further impacts are possible.



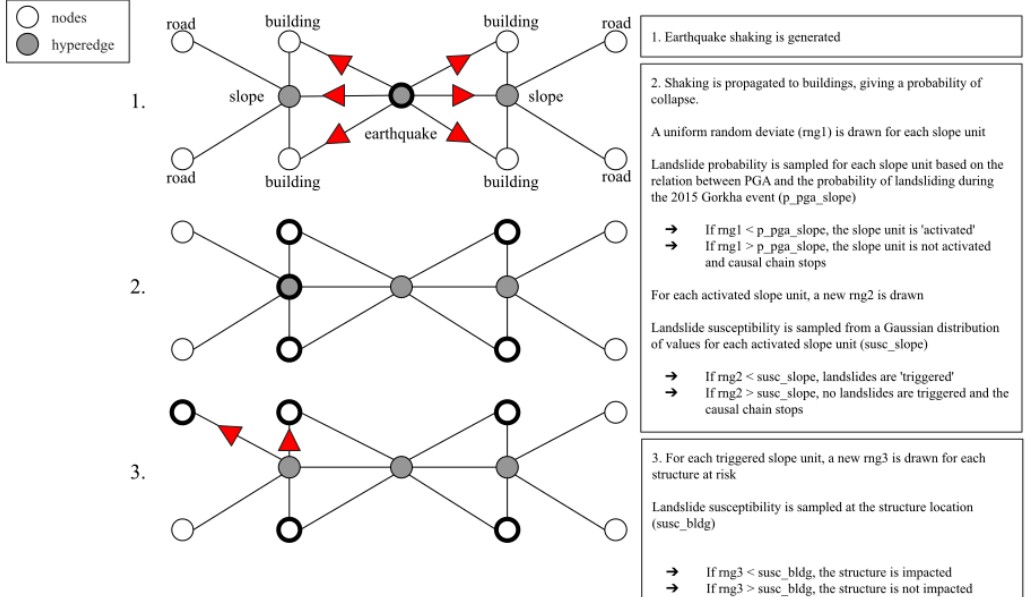

259

**Figure 4. Step-by-step overview of the hypergraph framework for modelling cascading multi-hazard impacts. The hypergraph is represented in a simplified example on the left and the algorithm steps are specified on the right. The simplified hypergraph assumes a landscape with two slope units, each of which contains two buildings and two road segments. The causal cascades of the algorithm are represented in three steps; from top to bottom, these are (1) earthquake shaking, (2) tests for 'activation' of a hillslope and 'triggering' of landslides, and (3) tests for impacts on structures by landslides. In the simplified hypergraph, black outlines show the hyperedges where hazards occur (e.g., landslides are triggered by the earthquake), and the nodes that are damaged by either shaking (step 2) or landsliding (step 3). The process is embedded in an iterative Monte Carlo simulation to determine the uncertainty associated with each step, creating a series of disaster scenarios that can be queried for further analysis.**


In the experiments shown here, the first step is to work through the hyperedge that connects the earthquake to the
individual buildings to assess their damage state. For each building, we assign the PGA value at the centroid of its 90 x
90 m METEOR grid cell. We use the high, middle, and low weighted mean fragility functions for that grid cell to
determine the likelihood of that building being completely damaged – which is equivalent to the proportion of buildings
within that 90 x 90 m grid cell in the METEOR dataset that is completely damaged. This likelihood of complete damage
[0,1], reproduces the weighted mean fragility when applied over the METEOR grid cell. The low, middle, and high
cases provide a range of outcomes for an individual building at a specific PGA value. The per-building likelihoods of
complete damage under the three cases can then be summed by slope unit or administrative area to give the total
predicted number of completely-damaged buildings in each area.

Next, we assess which slope units are 'activated' by ground shaking (Fig. 4). Activation of a slope unit means that the
ground accelerations are high enough to potentially trigger one or more landslides, if this is permitted by the
topographic conditions as represented by the landslide susceptibility. Again, this allows the stimulus to propagate within
the earthquake hyperedge to the slope unit, and potentially to cascade within that slope unit (and affect buildings or road
segments within it). In these experiments, we conduct a logistic regression between PGA and the locations of landslides
in the inventory of coseismic landslides triggered by the 2015 Gorkha earthquake (Kincey et al., 2021) to define the



regional-scale probability of landslide occurrence as a lognormal function of PGA (see Supplemental Information and
Fig. S3). We calculate the mean PGA value within each slope unit, and use that to determine the corresponding
probability of landsliding within the slope unit from the lognormal function. That probability, in turn, is compared with
a uniform random deviate to determine whether each slope unit is activated or not. Thus, over large numbers of
simulations, slope units with more observed coseismic landslides will be activated more frequently, but the exact pattern
of activations in each individual simulation – and thus the portion of the hypergraph network that is sampled – will vary.

For all slope units that are activated, the model proceeds to subsequent hyperedges to assess whether buildings or road
segments are affected by direct landslide occurrence (Fig. 4). In the experiments shown here, this is a two-step process.
We first check if a landslide occurred within the slope unit. Even if the shaking was strong enough to potentially trigger
a landslide (i.e., the slope unit was 'activated'), it might still have a low likelihood of experiencing landsliding due to
low susceptibility (i.e., it was not 'triggered'). Triggering in the slope unit is determined by drawing a value (A) from a
Gaussian distribution of landslide susceptibility with the same mean and standard deviation as the distribution of
susceptibility values in that slope unit, and comparing that value with a uniform random deviate (B). We employ a
Gaussian distribution for efficiency, as this can be calculated in advance of the simulation, and note that it provides a
reasonable fit to the actual distribution across a wide range of slope units (Supplemental Information, Fig. S4). If the
susceptibility value A is smaller than B, then no landslide has occurred in that slope unit, and propagation along that
hyperedge stops. If A is larger than B, then one or more landslides has occurred in that slope unit. We then check if each
building and road segment within the slope unit is affected by this landsliding by comparing the landslide susceptibility
value at the infrastructure location with another uniform random deviate. If the random deviate exceeds the landslide
susceptibility value, then the building or road segment remains unaffected by the landslide (in other words, even if a
landslide happens in the slope unit, it doesn't affect the building or road). Then, the simulation continues to evaluate
other buildings or roads within the same slope unit, and then moves on to other slope units activated by the earthquake.
If the random deviate is less than the susceptibility value, then the building or road segment is impacted by landsliding.
In this case, we add it to the pool of affected elements for this simulation and move to the next building or road. We
continue this process to search iteratively through all slope units in the network to generate a single cascading impact
scenario.

### 3.3 Outputs and evaluation


The iterative simulation process outlined above is repeated within a Monte Carlo framework to create an ensemble of
scenarios, each of which explores a different set of outcomes within the same set of hyperedges. In the experiments
shown here, we generate 10,000 scenarios from the initial stimulus of the 2015 Gorkha earthquake. Hence, all scenarios
in these experiments use the same spatial distribution of PGA values and thus the probability of an individual building
suffering complete damage by shaking stays the same. What differs between scenarios are the details of which slope
units are activated, which slope units experience landsliding, and which buildings or road segments are impacted by
those landslides. Thus, we take the likelihood of a structure being affected by landsliding over the whole ensemble as
the proportion of the 10,000 scenarios in which the structure is impacted. This leads to a shaking impact likelihood and
a landslide impact likelihood, both in the range [0,1], for each of the buildings and road segments in our combined
dataset.



To explore the trade-off between spatial resolution and model performance, we aggregate the structure-level results over
successively larger administrative units. Nepal is divided, from smallest unit to largest, into 6,743 wards, 753 urban and
rural municipalities, 77 districts, and 7 provinces. Aggregation across these units allows us to evaluate the performance
of the model against independent measures of earthquake impacts from the 2015 Gorkha earthquake at different spatial
resolutions. For buildings damaged by earthquake shaking, we evaluate the model in two ways. First, we sum up the
per-building likelihoods of complete damage in each district for the low, middle, and high fragility estimates – which
yields the number of completely-damaged buildings in each case – and compare those sums to incident reports
summarising the number of "fully damaged" buildings per district and published on the Government of Nepal's Bipad
Portal (http://drrportal.gov.np/ – see also Chaulagain et al., 2018) based on the Post-Disaster Damage and Needs
Assessment (PDNA) (National Planning Commission, 2015). This assesses the ability of the model to estimate the
absolute number of damaged buildings. While this data remains the most extensive for validation purpose, the PDNA
was done urgently after the disaster with limited systematic gathering hence it relies on judgement by the PDNA
participants and, therefore, carry significant uncertainty (Lallemant et al., 2017). Note that wards and municipalities
were defined in the federal restructuring of Nepal in 2017, and so data on damaged buildings from the 2015 earthquake
are not available at ward or municipality level. Second, we take the mean likelihood of complete damage in each
district, in the range [0,1], and compare that with the presence or absence of damaged buildings in each of the 77
districts. This second measure is independent of the absolute number of buildings, and gives information instead on the
ability of the model to anticipate the occurrence of one or more completely damaged buildings in an area.

For structures impacted by landslides, we derive similar statistical measures for model evaluation. First, we sum up the
per-structure likelihoods of landslide impact over successively larger areas of aggregation – ward, municipality, district,
and province. Because there are no systematic published data on observed landslide impacts on buildings and roads in
the 2015 earthquake, we generate an estimate of affected structures by overlaying the coseismic landslide polygons
from Kincey et al. (2021) on our building and road dataset; all structures that intersect with a mapped landslide polygon
are assumed to have been impacted by landsliding in the earthquake. Note that this measure of landslide impacts does
not consider the significant post-earthquake changes in landslide footprint and debris runout (e.g., Tian et al., 2020;
Kincey et al., 2022). Also, the coseismic landslides were mapped on medium-resolution satellite imagery (c. 10 m,
equivalent to our DEM and derived topographic metrics) and so will have omitted small landslides or rockfalls,
especially in areas of dense vegetation or steep topography (e.g., Williams et al., 2018); this error and the inherent
uncertainty in mapped landslide outlines (Kincey et al., 2021) mean that our estimate of the number of
landslide-affected structures is likely to represent a lower bound. We then sum the observed number of impacted
buildings and road segments by administrative area to compare with our modelled totals. We also compare the mean
likelihood of landslide impact, averaged by administrative area and ranging from [0,1], with the presence or absence of
landslide impacts in that area. We evaluate the relationship between these parameters with the area under the ROC
curve and the F1 score.

**4. Results**

**4.1 Impacts from earthquake shaking**

We first consider modelled impacts from earthquake shaking alone. Unsurprisingly, the probability of complete damage
per building, or equivalently the proportion of completely-damaged buildings within each 90 x 90 m exposure grid cell,



closely matches the estimated PGA contours from the Gorkha earthquake (Fig. 5A). There are particularly high
probabilities in the hill and mountain districts, especially to the east and northeast of Kathmandu, where the values
exceed 0.7. Notably, these values generally increase to the north and this increase is cut off only by the lack of buildings
above elevations of around 3,500 m in northern Nepal (visible as the white areas in Fig. 5A). The Kathmandu Valley
itself yields a low proportion of completely-damaged buildings, despite moderately high PGA values, due to the
preponderance of less-fragile building types.

We convert the proportion of completely-damaged buildings per grid cell into a sum total aggregated over
municipalities (Fig. 5B) and districts (Fig. 5C). These totals reflect the PGA pattern and the weighted mean fragility
functions, but importantly also the number of buildings within each administrative area. When aggregated by
municipality, the largest modelled totals tend to occur in the more densely-populated Middle Hills in the vicinity of
Kathmandu, rather than the more sparsely-populated north. There are some notable exceptions to this pattern, such as
Bharatpur to the south of the earthquake epicentre (Fig. 5B), which combines a large stock of fragile building types with
moderately high PGA values. When aggregated by district, the largest modelled totals are again dominated by areas
with both large numbers of buildings and moderate to high PGA values (Fig. 5C). With the exception of Chitwan to the
south of the epicentre, the largest totals are found in districts where PGA exceeded 0.4 g. It is instructive to compare the
aggregated pattern by district to the actual numbers of completely-damaged buildings (Fig. 5D). There are broad
similarities between modelled and observed totals, especially in the hill and mountain districts of Sindhupalchok,
Nuwakot, and Kavrepalanchok. Notably, the model over-predicts the impacts in districts close to the epicentre,
including Gorkha and Chitwan, and under-predicts the impacts at the eastern margin of the rupture in Dolakha (Fig.
5D).





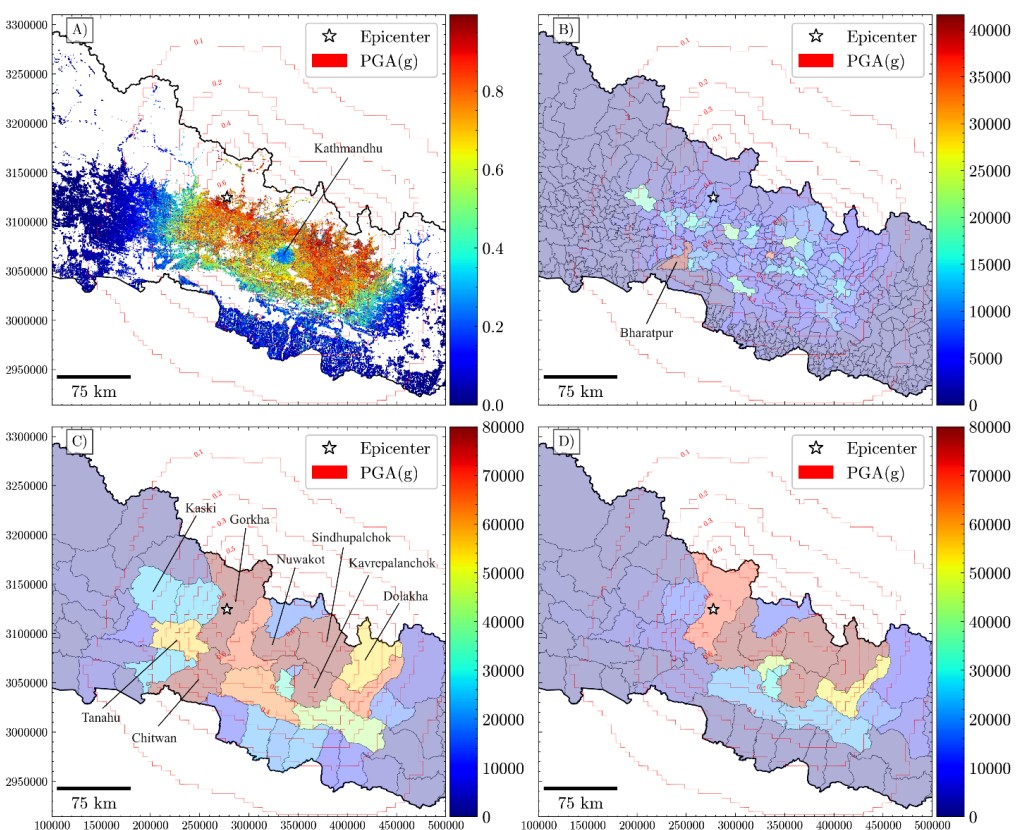


**Figure 5: Modelled building impacts from shaking in the 2015 Gorkha earthquake. In all panels, the red contours show the estimated PGA values from the earthquake in g. Note that these results are derived from the middle-case fragility functions in Fig. 4. A, modelled probability of complete damage for individual buildings across the country. This is equivalent to the proportion of completely-damaged buildings in each 90 x 90 m grid cell in the METEOR exposure dataset. B, modelled sum total of completely-damaged buildings aggregated by municipality. C, modelled sum total of completely-damaged buildings aggregated by district. D, actual sum of reported "fully damaged" buildings aggregated by district. Note similar colour scales in panels C and D.**

399

To better visualise the agreement between modelled and observed totals of completely-damaged buildings, we compare the observed totals for all 77 districts in Nepal with model results using the high, middle, and low fragility cases (Fig. 6A). For most districts with non-zero impacts, the observed totals fall within the range of model results using the different fragility curves, with a slight bias toward model over-prediction (Fig. 6B). Among the top 15 districts in terms of modelled impacts, observed impacts fall below that range in three districts (Chitwan, Tanahu, and Kaski; see Fig. 5C for locations), within that range in 11, and above that range in only one (Dolakha). Alternatively, out of the '14 worst-affected districts' identified by the Government of Nepal, observed impacts fall within the range of model results in thirteen districts, with Dolakha being the only outlier. The model thus appears to be somewhat conservative in that it slightly over-predicts building impacts due to shaking in the 2015 earthquake. The mismatch between modelled and observed totals is not clearly related to building typologies (Fig. 6C). There may be a weak correlation with shaking;





districts with significant over-prediction tend to be those with lower mean PGA values (typically <0.44 g) while
Dolakha has a larger mean PGA (0.59 g), and we explore this point in the Discussion.

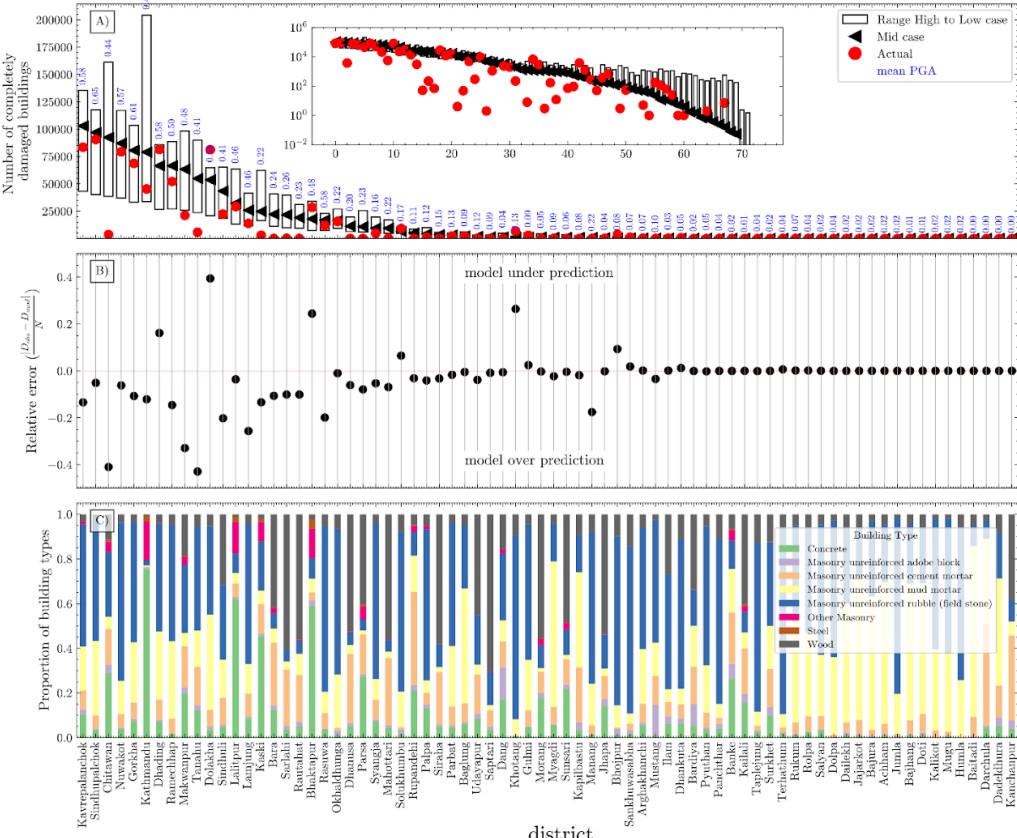


**Figure 6: A, comparison of modelled and observed numbers of completely-damaged buildings per district in the 2015 Gorkha earthquake. Bars show the range of modelled results for each district using high and low fragility cases (see Fig. 4), with the middle case shown by the black arrow. Red dots show the reported numbers of "fully damaged" buildings. Blue numbers show the mean PGA for each district, in g. The inset shows the same quantities with a logarithmic y-axis scale. B, mismatch between observed ($D_{obs}$) and modelled ($D_{mod}$) numbers for each district, normalised by the total number of buildings in that district ($N$). Negative values indicate model over-prediction, while positive values indicate model under-prediction. Note that impacts in most of the districts with non-zero damage values are slightly under-predicted. C, proportion of different building types in each district from the METEOR exposure data set. There is no clear correlation between the residuals in panel B and the dominant building types.**


## 4.2 Impacts from coseismic landslides


As with shaking damage, the modelled probability of a building (Fig. 7A) or road segment (Fig. 7D) being impacted by
a coseismic landslide scales with PGA; this is simply a consequence of the assumed relationship between PGA and
landslide triggering (Fig. S3). Higher probability values are found in northern areas of Nepal, where landslide



susceptibility is elevated (Fig. S2). We aggregate these probabilities to estimate the number of impacted buildings and
road segments at the municipality (Fig. 7B, E) and district (Fig. 7C, F) levels. The regions experiencing the highest
predicted impacts closely align with those observed, notably concentrated in Sindhupalchok district, where both
modelled and observed landslide impacts are most prevalent (Fig. 7C, F). Again, these areas predominantly lie in
northern Nepal where susceptibility to landslides is greatest, contrasting somewhat with the distribution of modelled
shaking damage. This disparity may stem from the higher and more widely dispersed density of buildings in the
southern regions. Consequently, while shaking-related damage appears diffuse, landslide-related damage is more
focused in specific regions due to localized exposure. Importantly, the model anticipates approximately an order of
magnitude fewer building impacts from landslides as compared to those damaged by shaking (note the scale difference
between Figs. 5 and 7). We also note that, while the overall spatial patterns of modelled building and road impacts are
similar, the model predicts somewhat higher numbers of road impacts (by about 50%), and that this generally matches
the observed differences in intersections between these infrastructure types with coseismic landslides (Fig. 7). Roads are
typically sited along or near valley floors , thus increasing their exposure to landslides. Additionally, there is a
significant association between roads and landslides (e.g., Hearn and Shakya, 2017; McAdoo et al., 2018), suggesting
that the interaction between landslides and roads may cover a broader spatial extent compared to the relationship
between landslides and buildings.







**Figure 7: Modelled structural impacts from coseismic landslides in the 2015 Gorkha earthquake. In all panels, the red**
**contours show the estimated PGA values from the earthquake in g. The red crosses show observed landslide impacts on**
**buildings (left column) and road segments (right column), derived by mapping the intersections between those structure**
**locations and the coseismic landslide inventory of Kincey et al. (2021). A, modelled probability of impact for individual**
**buildings across the country. B, sum of per-building probabilities aggregated by municipality, of which there are 753 in**
**Nepal. C, sum of per-building probabilities aggregated by district, of which there are 77 in Nepal. D, modelled probability of**
**impact for individual 100 m road segments across the country. E, sum of per-road segment probabilities aggregated by**
**municipality. F, sum of per-road segment probabilities aggregated by district.**

The correlation between the modelled and observed numbers of buildings impacted by landslides depends upon the area
over which they are aggregated (Fig. 8). At province (n = 7) and district (n = 77) levels, there is an approximately linear
relationship between modelled and observed numbers of buildings, with a Pearson's correlation coefficient >0.80 (Fig.
8). At municipality and ward levels, however, the correlation is much weaker. Notably, modelled numbers of buildings
over-predict the observed totals by a factor of about 50-100, irrespective of the administrative area. Similar results are
seen for road segments: good linear correlations for province- and district-level aggregation, much weaker performance
for municipalities and wards, and over-prediction of impacts by a factor of about 20-25 (Fig. 8).





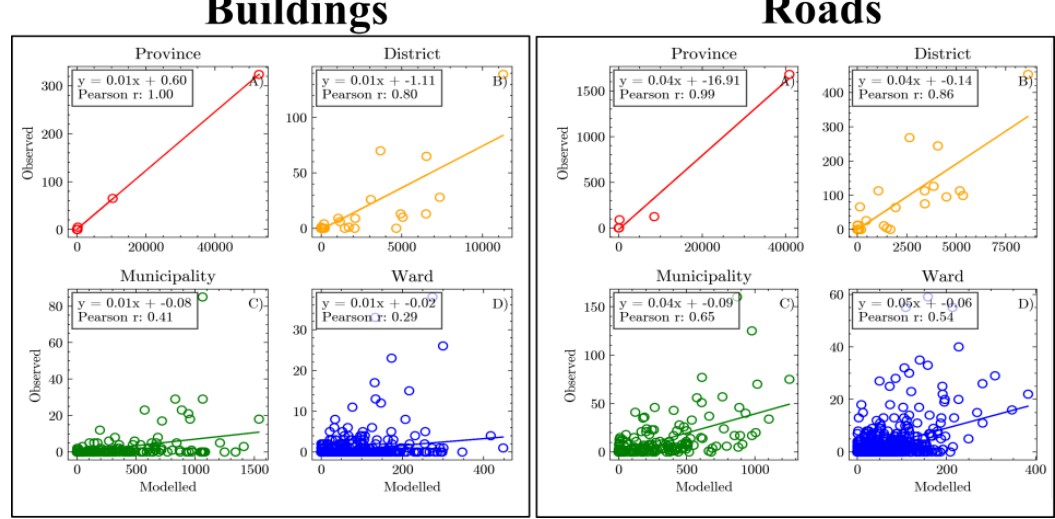


**Figure 8: Comparison of modelled (x-axis) and observed (y-axis) numbers of building and road impacts from coseismic landslides in the 2015 Gorkha earthquake, summed over different administrative areas. Straight lines show best-fit linear regression results. Note differences in axis limits depending on the area of aggregation by province (red), district (orange), municipality (green), or ward (blue).**

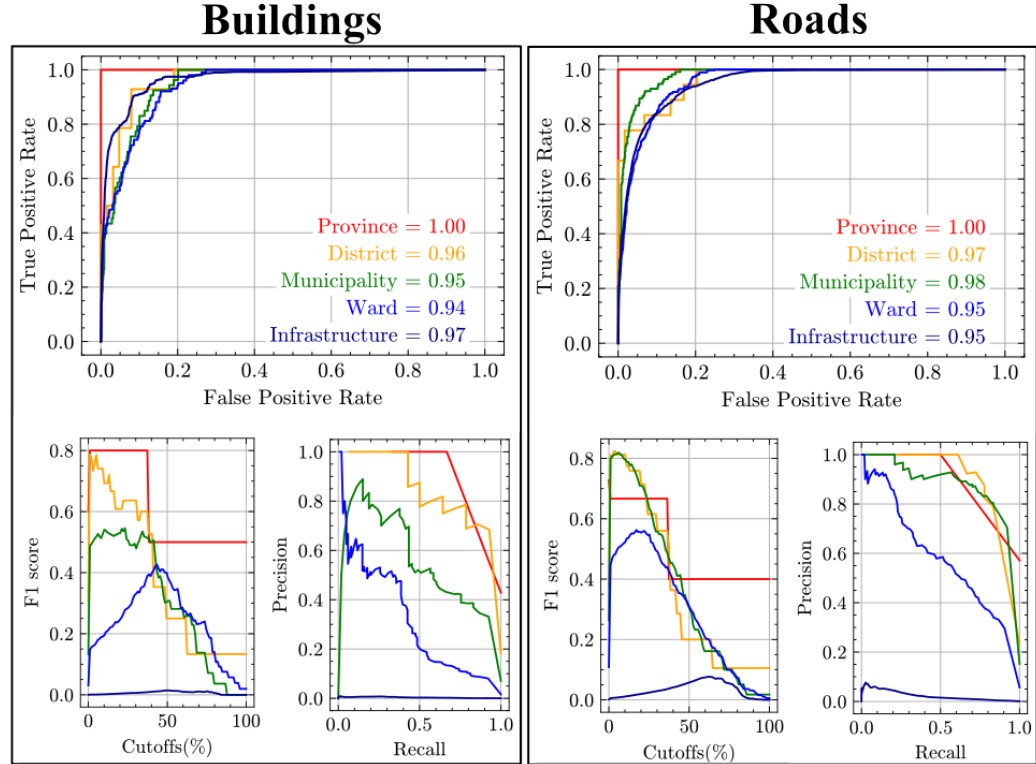

469



**Figure 9: ROC (top), F1 (lower left), and precision-recall (lower right) curves for coseismic landslide impacts of buildings and road segments aggregated over province, district, municipality, ward and at the individual infrastructure scale. Numbers in the top panels show the area under the ROC curves. Line colours match the symbol colours in Fig. 8.**

As a more permissive test of the model's ability to anticipate landslide impacts, we also compare the mean likelihood of landslide impacts, averaged by administrative area, with the presence or absence of impacts in those areas. While the area under the ROC curves is high for all aggregation levels (Fig. 9), this is likely due to the strong imbalance between prediction categories (i.e., there are many more non-impacted buildings than impacted buildings, so the ROC curve is dominated by the large number of true negative model results). In contrast, precision-recall curves show a progressive decrease in model performance at progressively smaller levels of aggregation, from province to ward, and very low precision at the scale of an individual building or road segment (Fig. 9). Because F1 scores combine precision and recall, they show a similar pattern (Fig. 9); across the full range of thresholds, F1 scores for both buildings and roads (Fig. 9) are highest for province- and district-level aggregation and lowest for ward-level aggregation. For an optimal model threshold, province-level aggregation achieves maximum F1 scores of around c. 0.8 for buildings and c. 0.65 for roads. The maximum F1 scores for buildings are also around 0.8 for districts and diminish progressively to 0.55 for municipalities and 0.4 for wards. For roads, the maximum F1 scores are 0.8 for districts and municipalities, and 0.55 for wards. In sum, these results indicate that, while the model can reproduce the spatial pattern of landslide impacts at the provincial or district scale, its predictive capability is much weaker when assessing impacts within smaller administrative units like municipalities and wards, and it should not be used to predict impacts to individual buildings or road segments.

## 5. Discussion

### 5.1 General observations

Overall, the hyperedge model is able to reproduce the overall spatial pattern of the impacts from the Gorkha earthquake. This lends some confidence that the model framework could be adapted to estimate the potential impacts from a future event, such as a large earthquake or rainstorm. While the computational efficiency of the hyperedge approach is a notable strength – enabling rapid simulations involving extensive elements, such as the approximately 7.1 million individual buildings and 3 million road segments in our case – its significance extends beyond speed and flexibility because it fosters the generation of multi-hazard scenario ensembles, diverging from the limitation of focusing solely on deterministic impact scenarios. Robinson et al. (2018) demonstrated the advantages of scenario ensembles over the more common approach of single deterministic scenarios, especially as a tool for facilitating awareness of what could be possible in a future event. While creation of multi-hazard scenario ensembles is our wider goal, the experiments shown here focus on multiple realisations of the same past event for the purpose of evaluation.

A key finding of the experiments is the trade-off between model performance, in terms of the ability to anticipate both the spatial pattern and number of impacts, and the resolution of the model outputs. Because of the probabilistic nature of the model and limitations in our understanding of exposure, earthquake shaking, and landslide susceptibilities, we cannot say with confidence which buildings were impacted by hazards related to the 2015 earthquake. As we aggregate the model results over increasingly large areas, however, our ability to rank those areas in terms of impact, and to estimate the number of structures affected, increases monotonically. While our results can therefore not be used to



anticipate the risk to individual households, they could be used by organisations working at a larger scale to identify
areas that are more or less prone to different types of hazards, and provide a relative ranking in terms of the number and
scale of expected impacts. Thus, the value and potential usefulness of the hypergraph approach as implemented here lies
more in informing planning over larger spatial scales, at which the model performs best, as opposed to rapid response to
a particular event where detailed spatial information would be required. There is some indication that absolute numbers
of affected structures could be generated for larger administrative units by extrapolating the scaling by our analysis of
the 2015 earthquake (see, for example, Fig. 8), but we hesitate to draw conclusions from a single earthquake without
further testing.

**5.2 Over-prediction and relative impacts between hazards**

We note that the model over-predicts the number of impacts at all levels of aggregation, and is therefore conservative in
terms of anticipating the scale of impacts for the 2015 earthquake. The possible reasons for this over-prediction are
likely to differ for shaking and landslide impacts. The mismatch in the number of buildings damaged by shaking is
especially notable for districts with moderate mean PGA values (typically <0.5 g; Fig. 6A). The sigmoidal fragility
functions used in the model are steepest at moderate PGA values (Fig. 3); for the middle case, this corresponds to PGA
values of ~0.2-0.5 g for the most common building types in Nepal. Thus, small uncertainties in PGA will yield large
differences in the likelihood of complete damage, and thus in the numbers of completely-damaged buildings in our
model experiments. This issue is compounded by the highly-uncertain values of ground motion in the Gorkha
earthquake stemming from the paucity of strong-motion recordings, as noted by Goda et al. (2015). We also note that
our experiments do not account for aftershocks, including the $M_w$ 7.3 earthquake that occurred on 12 May and that
ruptured the eastern end of the 25 April slip patch under Dolakha district (Avouac et al., 2015). This event likely led to
additional building damage which was included in the observations but is not simulated here, perhaps leading to
under-prediction in Dolakha in particular.

Over-prediction of observed landslide impacts, in contrast, may result from a range of different factors. As noted above,
in the absence of an independent dataset of landslide impacts on buildings or roads in the 2015 earthquake, we have
generated these data by intersecting those elements at risk with the coseismic landslide inventory of Kincey et al.
(2021). This is likely to underpredict the actual number of impacts due to errors and limitations in landslide mapping as
well as the potential for buildings to be omitted from the Humanitarian OpenStreetMap database. It is also important to
note that our approach relies on a probabilistic sampling of an underlying landslide susceptibility dataset in order to
anticipate (1) the slope units in which a landslide is most likely to be triggered, and (2) the buildings and road segments
that were most likely to be affected. Our results are thus highly dependent upon the quality of the underlying
susceptibility information. In the experiments described here, susceptibility is a static quantity that depends only upon
local topography. Because we are focused on a single event, there is no direct provision for dynamic variation in
susceptibility over time or for other factors that may affect landslide occurrence, such as the presence or absence of
antecedent rainfall, soil moisture or other measures of ground condition, or land cover. Further applications of the
model could incorporate susceptibility estimates that are trained on other landslide inventories – for example,
time-varying susceptibility that captures the evolution of landslide hazard over time (e.g., Tian et al., 2020; Kincey et
al., 2021, 2022) or that depends upon other causative factors (e.g., Reichenbach et al., 2018).



Our model result that the number of buildings damaged by ground shaking is approximately an order of magnitude
greater than that impacted by landslides is difficult to test directly because of the lack of a systematic description of the
sources of building damage in the 2015 Gorkha earthquake. It is broadly consistent, however, with previous work on the
relative importance of secondary hazards – including landslides – and ground shaking in determining earthquake losses.
Bird and Bommer (2004) assessed the relative impacts of ground shaking and ground failure on direct and indirect
losses in earthquakes. They found that fatal landslides occurred in 10 of their 50 studied earthquakes and that landslides
could be the primary cause of building damage in affected areas, locally overshadowing ground shaking. Overall,
however, ground shaking was the primary cause of building damage in 88% of their studied earthquakes, and landslides
in only 6%. They also found that landslide-induced disruption of road or transport networks was much more common
than building damage, which matches our model results for the Gorkha earthquake. Daniell et al. (2017) argued that
ground shaking has caused 62% of total economic costs in earthquakes over the period 1900-2016, with landslides
responsible for 5% of total costs. Marano et al. (2010) found that 21.5% of the fatal earthquakes in the PAGER-CAT
database had deaths due to secondary hazards, but that these were rarely the main cause of death. Landslides were the
leading cause of non-shaking-related deaths if the 2004 Great Sumatra earthquake was excluded, although they
accounted for about an order of magnitude fewer deaths than ground shaking. In contrast, Budimir et al. (2014)
demonstrated that earthquakes with landslides typically cause more fatalities than those without, independent of other
factors such as earthquake size or affected population. Their results demonstrate the need to account for the full
multi-hazard cascade in anticipating losses at anything other than a simplified regional scale (e.g., Bird and Bommer,
2004; Daniell et al., 2017).

**5.3 Limitations**

While the model operates on a hyperedge that connects every structure within the dataset, there are a number of factors
that cannot be resolved at a building scale. Notably, PGA values were gridded at a spatial resolution of 100 by 100 m,
meaning that we have no information on the actual accelerations experienced by individual buildings or road segments.
Similarly, while landslide susceptibility was estimated using a comparatively fine-scale DEM with a grid size of 10 x 10
m, each individual building or road segment occupies at most a few grid cells and the susceptibility values are thus
highly location-dependent. It is also important to note that we do not simulate the triggering, occurrence, and runout of
individual landslides, nor do we 'place' landslides in the landscape as would be done for example in a landscape
evolution model (e.g., Croissant et al., 2017; 2019). Such a calculation would dramatically increase both the model
complexity, making it infeasible to construct a multi-hazard scenario ensemble at a national scale. Because of this
limitation, we cannot directly evaluate which elements at risk are directly impacted by landslides, nor can we anticipate
which elements may be affected by remobilisation and runout of landslide debris (e.g., Kincey et al., 2022). By
sampling the landslide susceptibility distribution for each slope unit, and the landslide susceptibility values for each
building, we are (over enough iterations) recovering those distributions, but we cannot overcome the inherent
uncertainty in susceptibility at those locations. Finally, the METEOR exposure dataset contains information on the
building types and numbers within each 90 x 90 m grid cell, but we have no information on the type and fragility of
individual buildings. Therefore, while impact likelihood is calculated at the scale of individual structures, we stress that
this estimate is only meaningful across the whole scenario ensemble, and should never be interpreted as a statement that
'building X will be affected by this earthquake'.



### 5.4 Other applications

Because of its efficiency, the framework allows exploration of other elements of model performance, including the distinction between false positive and false negative errors. While performance measures such as the area under an ROC or precision-recall curve can be used to define an 'optimum' model outcome, the model application and users may determine which type of error is more important to minimise. For example, a humanitarian organisation may view false positives as more acceptable than false negatives; the former may lead at worst to unnecessary preparations, whereas the latter means that impacts are not anticipated and may delay relief and recovery efforts. By quickly generating numerous multi-hazard scenarios, the framework can be run with users to explore these different outcomes, and to examine the specificity of model results to the details of a particular scenario (e.g., Robinson et al., 2018). The model could also be used to explore 'what-if' questions with users to examine the effects of particular interventions or remediation measures. In addition, the efficiency of the framework could be used to explore the evolution of risk over time, where increased simulation length or time resolution would lead to an increase in computational cost. Thus, the effects of policy decisions, climate change and consequent changes in hazard occurrence, or demographic shifts on the pattern of anticipated impacts could be explored (Zschau, 2017).

The flexibility of the hyperedge framework also lends itself to other types of simulation. Other elements of the multi-hazard chain shown in Fig. 2 could be included; for example, susceptibility to landslide debris remobilisation and runout could be included and sampled for each element at risk, allowing the model to anticipate both the direct impacts within an event as well as potential longer-term impacts arising from later secondary hazards (e.g., Fan et al., 2019; Kincey et al., 2022). Impacts from other types of driving events, such as monsoon rainfall, could also be explored. It would be feasible, for example, to generate an ensemble of scenarios around different rainfall patterns associated with a seasonal monsoon outlook, or with different iterations of shorter-term weather forecasts, to look at the pattern and specificity of impacts. Such an application would be subject to the comparatively low spatial resolution of both observational (e.g., Hou et al., 2014) and forecast rainfall data products, so that – just as with the earthquake scenarios developed here – the impact results at the scale of an individual structure would not be meaningful. The hyperedge framework would, however, allow exploration of the trade-offs between aggregation and model performance, as demonstrated here, and could be useful for informing humanitarian contingency planning for annual rainfall-related impacts in Nepal and other monsoon-affected countries.

### 6. Conclusions

Accounting for the multi-hazard aspects of risk is crucial for disaster risk reduction and humanitarian planning. Traditional approaches to risk modelling tend to omit the interactions between hazards and, even when these interactions are accounted for, may struggle to meet the computational demands posed by such complex scenarios. Here, we demonstrate that a new model based on hypergraph theory, a type of network modelling approach, is able to efficiently simulate multi-hazard risk. The model framework accounts for the interactions between a driving stimulus such as an earthquake or rainstorm with processes on the landscape (such as landslides) and exposed infrastructure. Beyond overcoming computational challenges, this framework can facilitate multi-hazard risk assessments by enabling the generation of ensembles to explore the importance of different geophysical hazards, larger areas, longer timeframes, and diverse counterfactual scenarios. This versatility enhances our understanding of complex risk landscapes and empowers decision-makers with valuable insights for proactive disaster preparedness and response strategies.




We explore the capabilities of the model through a case study of the 2015 $M_w$ 7.8 Gorkha earthquake in Nepal, which
caused widespread damage due to both primary shaking and secondary landslides. We find that the model can reproduce
the overall spatial pattern of earthquake impacts. The observed numbers of completely-damaged buildings in most
districts, including 13 out of the 14 worst-affected districts, fall within the range of model predictions, which depends
primarily on the assumed fragility functions for the typical building types found in Nepal. The model also broadly
reproduces the spatial patterns of structures that were damaged by coseismic landslides in the earthquake, although it
overestimates the absolute number of impacts. This may be due to limitations in the data used by the model to
determine impacts. Importantly, there is an increase in model performance when the results are aggregated over larger
administrative areas; the model does a reasonable job of anticipating the relative impacts at a province or district scale,
but performs much less well at the smaller scales of municipalities or wards. This result suggests that the hypergraph
framework could be usefully applied to rank administrative areas by expected impacts, for example due to a future
earthquake or rainstorm, to underpin pre-disaster contingency planning efforts where large-scale trends are more
important than detailed impact predictions. The computational efficiency of the hypergraph framework, even at the
scale of an entire country such as Nepal, lends itself to the generation of multiple impact scenarios and raises the
possibility of using an ensemble of potential scenario results rather than depending upon single-event scenarios for
disaster preparedness and planning.

**Author contributions**

Funding was acquired by ALD, TRR, and NJR. The study was conceived by ADu, TRR, ALD, and NJR. ADu wrote
the code and carried out the numerical experiments with input from TRR, ALD, NJR, RMR, and MEK. ADu and ALD
prepared the original draft of the manuscript and all authors contributed to review and editing.

**Competing interests**

The authors declare that they have no conflict of interest.

**Acknowledgements**

This research has been supported by the UK Global Challenges Research Fund through the NERC Multi-Hazard and
Systemic Risk programme (grant NE/T01038X/1). The AW3D DEM (©JAXA, RESTEC and NTTDATA) is licensed
via Durham University (UK), with funding from the DFID-UKRI SHEAR programme (project number 201844-112).



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
