# Peer review of "Impacts from cascading multi-hazards using hypergraphs: a case"

_EGUsphere, 2024_

## Author Comment (AC2)

**Impacts from cascading multi-hazards using hypergraphs: a case study from the 2015 Gorkha earthquake in Nepal**

Response to reviewers

Reviewers 2

Dear Authors,

I was invited to review your very interesting paper. The paper presents a hyperedge model approach for multi-hazard risk assessment, which is innovative and worth publishing. The main advantage of the method is the increased speed of calculation, and the possibility to generate many ensembles incorporating the uncertainty of the risk components.

Concerning the computational advantages of this model, I would have liked to see more information on the way of calculating using hypergraphs, in terms of software, platforms, and calculation speed.

**[Authors] The calculations were done on a MacBook Pro M1 2020 with Apple M1 chip 16GB RAM and 8 cores using Python. The model generates earthquake damage for 7.1 million buildings and 10000 landslide scenarios with impact on buildings and 3 million 100m segment of roads runs in about 15 minutes.**

The model is demonstrated on a national scale in Nepal, with the aim to evaluate the damage to individual buildings and road segments. Yet the input data is gridded, the analysis is done on slope units, and the specific exposure of individual buildings cannot be assessed, as well as the runout of landslides. It is not clear to me why you wanted to go to this building level for a national scale analysis.

**[Authors] The hyperedge methods required a conceptualisation of the system to be modelled as nodes and (hyper)edges. As the resolution of the landslide susceptibility is 10m, the impact of landslide could be computed at individual building scale. It provides us the advantage of having the freedom to scale up and assess the quality of the prediction at different aggregation levels.**

Whereas the hyperedge framework has a large potential for multi-hazard risk assessment, the multi-hazard modeling component in this paper is still rather modest. The only interaction that is considered is related to earthquake-induced landslides. Other possible follow-up cascading events, such as landslide dams, and their breakup, or debris flows are not

considered. The potential applicability of the proposed method for such more advanced interactions is mentioned in the discussion section, but not further worked out. This would be a nice topic for a follow-up publication.

**[Authors] We absolutely agree with this statement. This paper is an initial attempt of using an innovative framework for including actual cascading effects in the modelling. Using hypergraph is hazard agnostic, the logical follow up work would be to construct longer cascades.**

The paper demonstrates its applicability to simulate the earthquake damage to buildings, and the earthquake-induced landslide damage to buildings and roads for the 2015 Gorkha event. The model is trained using the 2015 Gorkha event landslide inventory, so it is in a way to be expected that the spatial patterns resemble the actual damage patterns.r4

Figure 1. Figure 1A is mostly a single hazard (although considering earthquake-induced landslides) and is also of rather poor quality. What if you would combine earthquakes with flooding, then the slope unit approach would not be appropriate? The concept of hypergraph is not clear from Figure 1b. Why are there three hypergraphs and not one or two in this example? What determines the number?

**[Authors] The image is 300 DPI so we will double check that the quality is not due to the pre-print.**

**The link between earthquake and flood could still be connected with slopes. The earthquake can activate slopes, which can trigger landslide in river segments that can then impact exposed elements through flooding (e.g. due to aggradation).**

**The incidence matrix and the connection patterns are specific to the purpose of the modelling. In this case, the cascading multi-hazard hyperedges are mapped out as interactions between earthquake, slope and assets (buildings and roads). The connections are, for now, reliant on expert opinions (e.g. we expect earthquake to have an impact on the slope and the slope to have an impact on the exposed elements).**

The method doesn't seem to analyse the direct damage due to ground shaking to roads. Why is that not considered?

**[Authors] Road damages from earthquakes are not reliably assessed from shakemaps but are rather caused by local fault displacements which locations and magnitudes are highly uncertain and was therefore not taken into account.**

Figure 2: the multi-hazard interactions when including rainfall are not addressed clearly in this figure. Rainfall-induced flooding will not be suitable to consider at a slope unit level.

Debris flows also may cover several units.  And the other multi-hazard interactions (e.g. landslide dams) and post-earthquake reactivation of landslides are not considered sufficiently in this figure. Also not that the exact interaction between landslides and infrastructure is not considered. Perhaps you could draw some hypothetical slope units with roads and buildings, and show which components are in the model and which are not?

**[Authors]  Indeed, flooding would necessitate a specific mapping of the interactions with the other processes. However, even if flooding is an important hazard in the Nepalese landscape, it was not in the mandate of the paper.**

As you mentioned in the discussion section the model might not work for debris flows that reach valleys, and to settlements and roads that are located at the outlet of valleys, as the model only considers slope units.

**[Authors] Indeed, we would need to consider how debris flow can be integrated and conceptualized into the graphical approach.**

It would be relevant to explain how the building dataset from METEOR was generated and how this could include the construction types and building values for the whole of Nepal. A description of the uncertainties involved would be helpful.

**[Authors] The METEOR project derived EO products to classify homogeneous regions characterized by differing levels of urbanization, including rural areas, residential neighbourhoods, urban centres, and industrial zones. These homogeneous regions, termed "development patterns," are associated with a "mapping scheme," which represents the distribution of structural attributes and profiles specific to each development pattern. The mapping schemes are developed by engineers who consult scholarly literature, building codes, and satellite/ground imagery to determine a country's traditional construction methods, common building materials, and engineering requirements. This information aids in estimating vulnerability classes and replacement costs. Structural distributions for each development pattern within a country are then constructed using nationwide census data in conjunction with satellite and ground-based observations (Kathmandu Living Labs for Nepal). (https://nora.nerc.ac.uk/id/eprint/533439/) – reference added to the manuscript. The capacity of the method to map out exposure was based on the city of Los Angeles where the datasets and census data are robust enough for validation for different levels of aggregation. The validation was done based on the comparison of the square footage from the EO protocol with the square footage from the census data, showing a good alignment. The vulnerability data, as far as I could find, was validated by cross referencing the EO mapping scheme with on-the-ground sampling of few thousands of buildings in Nepal.**

The fragility curves presented in Figure 3 show that even with extreme levels of ground shaking over 1.5 g many building types do not result in complete damage in the low Case (e.g. unreinforced fieldstone not reaching more than 60% probability of complete damage with PGA of 3 g). Unreinforced masonry with cement mortar seems to be more vulnerable than mud mortar in the higher probability range. This is also counterintuitive, and requires further explanation, as also the large deviation of the lower curve.

**[Authors] After a literature review of the different fragility functions available for Nepal, it was clear that large discrepancies existed in vulnerability estimations. Hence, the decision was made to include a High, Mid and Low case based on the upper and lower bound fragility functions found in scientific literature to compensate for the lack of consistency in vulnerability measures. Unreinforced masonry with mud mortar, in the mid case would reach complete 50% probability of building damage for a PGA of ~0.2g while, for unreinforced cement mortar it would take double the PGA value to achieve the same probability of damage.**

The use of a static susceptibility model which is trained on the landslide inventory caused by the Gorkha earthquake, without including the earthquake shaking as a causal factor, might be problematic.

The simulation takes quite a few shortcuts or assumptions, which are understandable given the limited data availability. The threshold used for defining the slope units with landslides based on the relation between PGA and landslide occurrence during the Gorkha earthquake doesn't take into account the terrain conditions.

**[Authors] The shaking value is included in the cascading chain as a triggering factor for the slope units based on the threshold values observed during Gorkha as described Lines 286-288. Earthquake shaking was discretized from topographic factors (contained in the 10m resolution susceptibility map) to allow the flexibility to vary earthquake scenarios.**

276-278: Can you explain how you can sum up the probabilities of the individual buildings per slope unit / administrative unit to obtain the number of destroyed buildings? E.g. if you have 100 buildings, each with a 50% probability of failure, do you then have 50 destroyed buildings?

**[Authors] Yes, the example you gave is the correct assumption.**

280-291 When you assess the relation between PGA and landslides on the basis of the Gorkha earthquake inventory, will the results then not mimic the situation of the Gorkha earthquake?

**[Authors] Yes, indeed. We realize that the argument is cyclical, but we were constrained by the data availability in Nepal.**

Also if the relation between PGA and landslides does not take into account other covariates, would this not have the effect that slope units that are not very steep but a high PGA are still "activated" ? the sentence "That probability, in turn, is compared with a uniform random deviate to determine whether each slope unit is activated or not" is not clear and could be explained better.

**[Authors] The landslide hazard is managed in three steps through the use of uniform random deviate. First, through a stochastic activation of the slope units if the PGA is large enough (i.e. the slope can potentially generate landslides). Second, through the sampling of the susceptibility distribution in the slope unit as a proxy for topographic covariate (i.e. the slope generate landslides). Third, sampling the susceptibility at the exposed element scale (i.e. is the exposed element impacted by a landslide runout).**

**It is possible to generate scenario where a high PGA would "activate" the slope unit but will not "trigger" landsliding because a flat topography point to a low probability of occurrence. We will add to the paper.**

295 "We first check if a landslide occurred within the slope unit" Why was this done? Is the analysis then not biased toward replicating the landslide inventory?

**[Authors] In the context of the multi-hazard algorithm, the sentence refers to the sampling of the susceptibility distribution of the slope units. If the value sampled exceeds a random uniform sample, a landslide is "occurring". As mentioned, the susceptibility model is based on existing inventory from the Gorkha earthquake, as few others are available, hence a bias exist toward the Gorkha event.**

299 How is this uniform random deviate (B) determined?

**[Authors] The value sample is coming from a uniformly distributed over the half-open interval [0, 1). In other words, any value within the given interval is equally likely to be drawn. This additional information has been added to the manuscript Line 300.**

303-305: if within a slope unit potentially landslides are triggered, how do you then determine how many buildings and roads would be impacted? Here again, you use a random value. The use of these random values in the method is not clear.

[Authors] for each building and road, a distribution of susceptibility values is available. From those individual distributions, a value is drawn. This value is then compared with a random uniform draw from 0 to 1. The purpose of this comparison is to allow the highest

susceptibility to mostly allow an impact but not always. This is a way to generate cascading effects that explore the uncertainty space and create different risk scenarios.

316-318: you create 10000 scenarios but these are all related to the Gorkha earthquake?

**[Authors] In this paper, the initial simulation is the Gorkha shaking indeed. The 10000 scenarios are not all identical because of the random value generations.**

387-389: What are the reasons for the over-and-under prediction? You are discussing these in the discussion section, but have you tried to reduce this overprediction by adjusting certain components?

**[Authors] This question was investigated and pointed to the PGA being the influencing factor as well as some typologies to a lesser extent. Below is a SHAP analysis of the relative error prediction at District levels. As Gorkha is our single reference point, we decided to present the raw results. For future work, a PGA cut off might be applied.**

[Figure]

Figure 5: the red contours are poorly visible. You might want to show them in a separate figure, and not repeat them in each map.

**[Authors] The Figures will be updated as per the comments of the other reviewer as well.**

Figure 6 is quite complex and is not focusing on the main topic of the paper. It is replicating single hazard earthquake building losses for a scenario earthquake. There are quite some outliers in this graph with large differences between observed and predicted. Isn't it logical that for most of the districts the results fall in between the low and high values?

**[Authors] Figure 6 is an output of the multi-hazard model as there is no distinct model assessing the earthquake and the landslide separately. The low and high values are the results of the weighted mean of the high case and low case for each building and their specific topologies aggregated at the district scale. Hence it is possible, due to the uncertainty in the fragility functions and shake map, that the model bounds events wouldn't accurately represent the actual event.**

Figure 7: The density of information in these maps is very high, and that makes them difficult to interpret. The actual landslide impacts are often overprinted by the other information, especially in A and D. PGA contours could be left out, and the color scale adjusted as most values are blue and it is not possible to see if red areas are due to the crosses or to the map values.

**[Authors] Figure 5 and 7 will be updated for clarity.**

459-462: could the weaker relation for earthquake-induced landslides not be due to the separation of the triggering PGA values to activate the slope unit, and the separate susceptibility values that were not considering PGA values?

**[Authors] From the USGS shake map, it seems that PGA values are not varying drastically over the slope units area most likely due to the lack of capacity to model high resolution amplification effect. Hence, the dynamic addition of PGA separately from the topographic variables is not expected to have large differences with the a joined susceptibility values.**

Figures 8 and 9: even though the overprediction of building damage is on the order of 50-100 times, and road damage of 20-25, the AUC values seem to be very good. Is this not caused by the many administrative units that didn't have damage at all? Or is it simply predicting damage/no damage per administrative unit? In that case, the figure is not so meaningful.

**[Authors] We believe that the AUC score, indeed high, are the results of the imbalance in the landslide impact, with many unaffected buildings and roads. To address this issue we used the F1 score as it takes into account both precision and recall, which are important metrics for imbalanced datasets.**

In the discussion, it is mentioned that the modeling was only done for the 2015 Gorkha earthquake that occurred on 25 April 2015, excluding the event of 12 May 2015. How did you separate the landslides caused by these two earthquakes? I understood that due to the fact that they occurred close to each other, the mapping of the co-seismic landslides could not differentiate well between the landslides caused by both events.

**[Authors] This topic was indeed discussed during the project but, as you mentioned, the two landslide inventories couldn't be differentiated and, therefore, an the "aggregated" landslide inventory combining the two events was used.**

For the modeling of earthquake-induced landslides it might be considered to apply Spatio-temporal data-driven modeling (e.g. Dahal, A., Tanyas, H., van Westen, C., van der Meijde, M., Mai, P. M., Huser, R. and Lombardo, L. (2024c) Space-time landslide hazard modeling via ensemble neural networks. Natural Hazards and Earth System Sciences 24(3), 823–845. It might be good to address this a bit more in the discussion section.

**[Authors] Thank you for the very interesting references. As the purpose of the paper was to develop a dynamic cascading framework across different hazards, we consider in this initial case study the immediate cascade from an earthquake and the subsequent damages to infrastructures. The temporality aspect and the reference you mentioned could potentially be combined with a graphical method to explore deeper cascading effects with improved susceptibility models included.**

In how far do the interpolated PGA values correctly represent the effect of topography, and would PGA values be the best predictor for landslide occurrence alone (See also: Dahal, A., Tanyaş, H. and Lombardo, L. (2024b) Full seismic waveform analysis combined with transformer neural networks improves coseismic landslide prediction. Communications Earth & Environment 5(1), 75.)

**[Authors] Thank you for the very interesting references. As the purpose of the paper was to develop a dynamic cascading framework across different hazards, we consider in this initial case study the immediate cascade from an earthquake and the subsequent damages to infrastructures. The temporality aspect and the reference you mentioned could potentially be combined with a graphical method to explore deeper cascading effects with improved susceptibility models included.**

The reference to Fan et al. 2019 is missing in the reference list.

**[Authors] Amended**

---

## Author Response (AR1)

**Impacts from cascading multi-hazards using hypergraphs: a case study from the 2015 Gorkha earthquake in Nepal**

Response to reviewers

Thank you for your constructive comments and the time you dedicated to reviewing the paper.

Reviewers 1

Dear Authors,

I was invited to review the Manuscript Number: egusphere-2024-1374 "Impacts from cascading multi-hazards using hypergraphs: a case study from the 2015 Gorkha earthquake in Nepal".

The use of graphs is certainly very interesting for exploring interactions between multi-hazards. The challenge of applying such a method on a large national scale is overcome by the proposed hypergraph approach. The main benefit derived is efficiency. The work is clearly and well-presented in its overall logic. The research question, case study, and methodology provide the necessary information to appreciate the approach in its generality. However, some passages are unclear and require minor additional information, as detailed below.

I would like to highlight a general aspect that I believe needs further clarification: are the limitations of this approach attributed to the use of hypergraphs or to the individual models used to model specific hazards (e.g., fragility functions, estimation of susceptibility maps, etc.)? In my understanding, the limitations are due to the choice of the latter. If so, I think it is important to clarify this in the discussion and propose alternatives for future implementations that could improve these limitations. Additionally, what are the advantages of using hypergraphs beyond the computational efficiency that makes them applicable on a large scale?

The graph methodologies in risk assessment allow, among other things, the analysis of graph topology to highlight potential systemic behaviour and impact propagation mechanisms (see as example ref 1). Is this possible with hypergraphs? I invite the authors to consider to discuss these potential applications or limitations. My question upon reading the novelty of the manuscript is whether hypergraphs are an innovative algorithm extending traditional risk of multi-hazard methodologies (beyond the multi-layer single hazard) to larger scales (thanks to their efficiency) or if they introduce a conceptually different approach to impact estimation? Please clarify this aspect in the discussion section

**[Authors] Thank you for your comments. The use of hypergraph has two main advantages, one as pointed out by the reviewer, was computationally efficient (even more efficient than a standard graphical framework, due to the "one to many" connections as described in the paper). The other fundamental advantage is the capacity of this framework to overcome the problem of combining hazard models of different nature for cascading multi-hazard risk assessments by simplifying and standardizing the structure of the initial model's datasets.**

**The limitations mentioned in section 5.3 Limitations are wide ranging in their domains, from structural engineering to the resolution of landslide susceptibility models. Academics are trying to overcome those limitations in their own rights. The purpose of the paper herein is to provide an innovative framework which allows cascading effects to be modelled while allowing any future progress across modelling domains to be captured.**

**In the paper herein, the hypergraph constitutes the backbone of the propagation algorithm (i.e. cascading scenarios) as it was initially realized in previous papers by the author using standard graphs (Dunant et al., 2021a, b) and typological measures were not the target of this study (Arosio et al., 2018 added Line 42 for reference on risk complexity). The typology of hypergraphs can indeed be studied with measures such as centrality (Hypergraphx and HyperG are examples of libraries). It might be interesting to dedicate a future paper to study various topological measures for longer cascading scenarios (e.g. added interactions landslide / river) and a "deeper" network of interactions.**

Detailed aspects include:

- Lines 284-292 are unclear; please reformulate with more explanations for clarity.

  **[Authors] The paragraph has been amended for clarity**

- The susceptibility section was too hasty, particularly the process of identifying "slope units" and the relationship between "slope units" and the extension of the landslide. Additionally, it is unclear whether the buildings and roads affected by landslides are only those falling within the "slope units" or if there is some estimation of the landslide's influence area. In either case, further explanation is needed.

  **[Authors] Additional information about the creation of the slope units has been added to the manuscript Lines 191-192 in addition to the Supplementary information material. Hopefully, the additional sentences also clarifies the landslide influence area, as it specifies "[…] focusing on whether buildings or road segments within the slope unit are directly affected by a landslide" Lines 311-312**

- A table summarizing the data used, specifying the main characteristics, including the different resolutions used could help the reading.

  **[Authors] all the data sources, resolutions and uses are describe as part of the flow of the paper and we believe that adding a table as supplementary material would therefore be redundant.**

- Figures 5 and 7 use a continuous scale for discrete colors, which is not intuitive. I suggest to explore other legend options.

  **[Authors] The figures have been updated for clarity.**

- The quality of the figures is low, which may be a pre-print issue. I suggest to check before the final version.

  **[Authors] All the figures have been saved with a DPI of 300 which should show a sharp image for the final version. We will double check.**

\*\*\*

**ref 1:** Arosio, M., Martina, M.L.V., Figueiredo, R., "*The whole is greater than the sum of its parts: A holistic graph-based assessment approach for natural hazard risk of complex systems.*", Natural Hazards and Earth System Sciences, 2020, 20(2), pp. 521–547

################################################################

Reviewers 2

Dear Authors,

I was invited to review your very interesting paper. The paper presents a hyperedge model approach for multi-hazard risk assessment, which is innovative and worth publishing. The main advantage of the method is the increased speed of calculation, and the possibility to generate many ensembles incorporating the uncertainty of the risk components.

Concerning the computational advantages of this model, I would have liked to see more information on the way of calculating using hypergraphs, in terms of software, platforms, and calculation speed.

**[Authors] The calculations were done on a MacBook Pro M1 2020 with Apple M1 chip 16GB RAM and 8 cores using Python. The model generates earthquake damage for 7.1 million buildings and 10000 landslide scenarios with impact on buildings and 3 million 100m segment of roads runs in about 15 minutes.**

The model is demonstrated on a national scale in Nepal, with the aim to evaluate the damage to individual buildings and road segments. Yet the input data is gridded, the analysis is done on slope units, and the specific exposure of individual buildings cannot be assessed, as well as the runout of landslides. It is not clear to me why you wanted to go to this building level for a national scale analysis.

**[Authors] The hyperedge methods required a conceptualisation of the system to be modelled as nodes and (hyper)edges. As the resolution of the landslide susceptibility is 10m, the impact of landslide could be computed at individual building scale. It provides us the advantage of having the freedom to scale up and assess the quality of the prediction at different aggregation levels.**

Whereas the hyperedge framework has a large potential for multi-hazard risk assessment, the multi-hazard modelling component in this paper is still rather modest. The only interaction that is considered is related to earthquake-induced landslides. Other possible follow-up cascading events, such as landslide dams, and their breakup, or debris flows are not considered. The potential applicability of the proposed method for such more advanced interactions is mentioned in the discussion section, but not further worked out. This would be a nice topic for a follow-up publication.

**[Authors] We agree with this statement. This paper is an initial attempt of using an innovative framework for including actual cascading effects in the modelling. As the hypergraph framework is hazard agnostic, the logical follow up work would be to construct longer cascades of hazards.**

The paper demonstrates its applicability to simulate the earthquake damage to buildings, and the earthquake-induced landslide damage to buildings and roads for the 2015 Gorkha event. The model is trained using the 2015 Gorkha event landslide inventory, so it is in a way to be expected that the spatial patterns resemble the actual damage patterns.

Figure 1. Figure 1A is mostly a single hazard (although considering earthquake-induced landslides) and is also of rather poor quality. What if you would combine earthquakes with flooding, then the slope unit approach would not be appropriate? The concept of hypergraph is not clear from Figure 1b. Why are there three hypergraphs and not one or two in this example? What determines the number?

**[Authors]**

1. **Image Quality: The image is 300 DPI so we will double check that the quality is not due to the pre-print.**

2. **Incorporating Flooding with Earthquakes: We agree that the combination of earthquakes and flooding presents a complex interaction scenario. However, even in such cases, the slope unit approach can still be relevant. For instance, an earthquake could destabilize slopes, potentially triggering landslides that impact river segments, leading to flooding downstream. This cascading effect could influence exposed elements such as buildings and roads due to aggradation in river channels. Thus, the slope unit remains an essential component in the hazard chain.**

3. **Clarification on Hypergraphs: The concept of hypergraphs in Figure 1B is designed to illustrate different levels of interactions among hazards and exposure elements. The three hyperedges (h1, h2, h3) represent distinct multi-hazard scenarios or cascading effects (e.g., earthquake and landslides) and their impact on specific assets (buildings, roads). The number of hyperedges is determined by the specific relationships being modelled. In this example, the interactions between earthquake, slope, and various assets are grouped into three distinct hyperedges based on expert judgment. The number of hyperedges could vary depending on the specific context and the complexity of the interactions being represented.**

The method doesn't seem to analyse the direct damage due to ground shaking to roads. Why is that not considered?

**[Authors] Road damages from earthquakes are not reliably assessed from shakemaps but are rather caused by local fault displacements. The locations and magnitudes of which are highly uncertain and was therefore not taken into account (Dunant et al., 2021b).**

Figure 2: the multi-hazard interactions when including rainfall are not addressed clearly in this figure. Rainfall-induced flooding will not be suitable to consider at a slope unit level. Debris flows also may cover several units. And the other multi-hazard interactions (e.g. landslide dams) and post-earthquake reactivation of landslides are not considered sufficiently in this figure. Also not that the exact interaction between landslides and infrastructure is not considered. Perhaps you could draw some hypothetical slope units with roads and buildings, and show which components are in the model and which are not?

**[Authors]**

**We have updated the figure accordingly to improve clarity and representation of the multi-hazard interactions.**

1. **Cascades from earthquake was the primary focus of this study, hence we chose to prioritize those hazards most relevant to the scope of our analysis. While flooding is indeed an important aspect of the Nepalese hazard landscape, it falls outside the specific scope of this study. However, potential alternative interaction schemes could be used to tackle hazards of different natures.**

2. **Debris Flows Across Slope Units: We have revised Figure 2 to better represent debris flows and their potential to span multiple slope units. This update now more accurately reflects the nature of debris flow hazards and their interactions across different units.**

3. **Landslide Dams: The updated figure also now includes a more explicit representation of landslide dams.**

4. **Interaction Between Landslides and Infrastructure: We have illustrated the interaction between landslides and critical infrastructure such as roads and buildings.**

**We have labelled these elements to distinguish between those included in our model and those that are not, thus clarifying the scope of our analysis.**

As you mentioned in the discussion section the model might not work for debris flows that reach valleys, and to settlements and roads that are located at the outlet of valleys, as the model only considers slope units.

**[Authors] Indeed, we would need to consider how debris flow can be integrated and conceptualized into the graphical approach. It might be interesting to conceptualize catchments in the hypergraph in addition to slope units.**

It would be relevant to explain how the building dataset from METEOR was generated and how this could include the construction types and building values for the whole of Nepal. A description of the uncertainties involved would be helpful.

**[Authors] The METEOR project derived EO products to classify homogeneous regions characterized by differing levels of urbanization, including rural areas, residential neighbourhoods, urban centres, and industrial zones. These homogeneous regions, termed "development patterns," are associated with a "mapping scheme," which represents the distribution of structural attributes and profiles specific to each development pattern. The mapping schemes are developed by engineers who consult scholarly literature, building codes, and satellite/ground imagery to determine a country's traditional construction methods, common building materials, and engineering requirements. This information aids in estimating vulnerability classes and replacement costs. Structural distributions for each development pattern within a country are then constructed using nationwide census data in conjunction with satellite and ground-based observations (Kathmandu Living Labs for Nepal). ([https://nora.nerc.ac.uk/id/eprint/533439/](https://nora.nerc.ac.uk/id/eprint/533439/)) – reference added to the manuscript Line 208-211.**

**The capacity of the method to map out exposure was based on the city of Los Angeles where the datasets and census data are robust enough for validation for different levels of aggregation. The validation was done based on the comparison of the square footage from the EO protocol with the square footage from the census data, showing a good alignment. The vulnerability data, as far as we could find, was validated by cross referencing the EO mapping scheme with on-the-ground sampling of few thousands of buildings in Nepal.**

The fragility curves presented in Figure 3 show that even with extreme levels of ground shaking over 1.5 g many building types do not result in complete damage in the low Case (e.g. unreinforced fieldstone not reaching more than 60% probability of complete damage with PGA of 3 g). Unreinforced masonry with cement mortar

seems to be more vulnerable than mud mortar in the higher probability range. This is also counterintuitive, and requires further explanation, as also the large deviation of the lower curve.

**[Authors] After a literature review of the different fragility functions available for Nepal, it was clear that large discrepancies existed in vulnerability estimations. Hence, the decision was made to include a High, Mid and Low case based on the upper and lower bound fragility functions found in scientific literature to compensate for the lack of consistency in vulnerability measures. Unreinforced masonry with mud mortar, in the Mid case would reach 50% probability of complete damage for a PGA of ~0.2g while, for unreinforced cement mortar it would take double the PGA value to achieve the same probability of damage.**

The use of a static susceptibility model which is trained on the landslide inventory caused by the Gorkha earthquake, without including the earthquake shaking as a causal factor, might be problematic.

The simulation takes quite a few shortcuts or assumptions, which are understandable given the limited data availability. The threshold used for defining the slope units with landslides based on the relation between PGA and landslide occurrence during the Gorkha earthquake doesn't take into account the terrain conditions.

**[Authors] The shaking value is included in the cascading chain as a triggering factor for the slope units based on the threshold values (i.e. logistic regression) observed during Gorkha as described Lines 286-288. Earthquake shaking was discretized from topographic factors (contained in the 10m resolution susceptibility map) to allow the flexibility to vary earthquake scenarios.**

276-278: Can you explain how you can sum up the probabilities of the individual buildings per slope unit / administrative unit to obtain the number of destroyed buildings? E.g. if you have 100 buildings, each with a 50% probability of failure, do you then have 50 destroyed buildings?

**[Authors] Yes, the example you gave is the correct assumption.**

280-291 When you assess the relation between PGA and landslides on the basis of the Gorkha earthquake inventory, will the results then not mimic the situation of the Gorkha earthquake?

**[Authors] Our analysis of the relationship between PGA and landslides is based solely on data from the Gorkha earthquake due to limited availability of comprehensive landslide inventories in Nepal. While this approach creates a circular argument, it represents the best available data for the region. Future research incorporating diverse landslide datasets from multiple earthquakes would strengthen our understanding of this relationship in the Himalayan context.**

Also if the relation between PGA and landslides does not take into account other covariates, would this not have the effect that slope units that are not very steep but a high PGA are still "activated" ?  the sentence "That probability, in turn, is compared with a uniform random deviate to determine whether each slope unit is activated or not" is not clear and could be explained better.

**[Authors] The landslide hazard is managed in three steps through the use of uniform random deviate. First, through a stochastic activation of the slope units if the PGA is large enough (i.e. the slope can potentially generate landslides). Second, through the sampling of the susceptibility distribution in the slope unit as a proxy for topographic covariate (i.e. the slope generate landslides). Third, sampling the susceptibility at the exposed element scale (i.e. is the exposed element impacted by a landslide runout).**

**It is possible to generate scenario where a high PGA would "activate" the slope unit but will not "trigger" landsliding because a flat topography points to a low probability of occurrence. We will add to the paper.**

295 "We first check if a landslide occurred within the slope unit" Why was this done? Is the analysis then not biased toward replicating the landslide inventory?

**[Authors] In the context of the multi-hazard algorithm, the sentence refers to the sampling of the susceptibility distribution of the slope units. If the value sampled exceeds a random uniform sample, a landslide is "occurring". As mentioned, the susceptibility model is based on the existing inventory from the Gorkha earthquake from Kincey et al. (2021), as few others are available, hence a bias exist toward the Gorkha event.**

299 How is this uniform random deviate (B) determined?

**[Authors] The value sample is coming from a uniformly distributed over the half-open interval [0, 1). In other words, any value within the given interval is equally likely to be drawn. This additional information has been added to the manuscript Line 303.**

303-305: if within a slope unit potentially landslides are triggered, how do you then determine how many buildings and roads would be impacted? Here again, you use a random value. The use of these random values in the method is not clear.

**[Authors] for each building and road, a distribution of susceptibility values is available. From those individual distributions, a value is drawn. This value is then compared with a random uniform draw from 0 to 1. The purpose of this comparison is to allow the highest susceptibility to mostly allow an impact but not always. This is a way to generate cascading effects that explore the uncertainty space and create different risk scenarios.**

316-318: you create 10000 scenarios but these are all related to the Gorkha earthquake?

**[Authors] In this paper, the initial simulation is the Gorkha shaking indeed. The 10000 scenarios are not all identical because of the random value generations mentioned earlier.**

387-389: What are the reasons for the over-and-under prediction? You are discussing these in the discussion section, but have you tried to reduce this overprediction by adjusting certain components?

**[Authors] This question was investigated and pointed to the PGA being the influencing factor as well as some typologies to a lesser extent. Below is a SHAP analysis of the relative error prediction at District levels. As Gorkha is our single reference point, we decided to present the raw results. For future work, a PGA cut off might be applied.**

[Figure]

Figure 5: the red contours are poorly visible. You might want to show them in a separate figure, and not repeat them in each map.

**[Authors] The Figures have been updated for clarity.**

Figure 6 is quite complex and is not focusing on the main topic of the paper. It is replicating single hazard earthquake building losses for a scenario earthquake. There are quite some outliers in this graph with large differences between observed and predicted. Isn't it logical that for most of the districts the results fall in between the low and high values?

**[Authors] Figure 6 is an output of the multi-hazard model as there is no distinct model assessing the earthquake and the landslide separately. The low and high values are the results of the weighted mean of the high case and low case for each building and their specific topologies aggregated at the district scale. Hence it is possible, due to the uncertainty in the fragility functions and shake map, that the model bounds wouldn't accurately contain the actual event.**

Figure 7: The density of information in these maps is very high, and that makes them difficult to interpret. The actual landslide impacts are often overprinted by the other information, especially in A and D. PGA contours could be left out, and the color scale adjusted as most values are blue and it is not possible to see if red areas are due to the crosses or to the map values.

**[Authors] Figure 5 and 7 have been updated for clarity.**

459-462: could the weaker relation for earthquake-induced landslides not be due to the separation of the triggering PGA values to activate the slope unit, and the separate susceptibility values that were not considering PGA values?

**[Authors] From the USGS shake map, it seems that PGA values are not varying drastically over the slope units area most likely due to the lack of capacity to model high resolution amplification effects. Hence, the dynamic addition of PGA separately from the topographic variables is not expected to have large differences with a joined susceptibility values.**

Figures 8 and 9: even though the overprediction of building damage is on the order of 50-100 times, and road damage of 20-25, the AUC values seem to be very good. Is this not caused by the many administrative units that didn't have damage at all? Or is it simply predicting damage/no damage per administrative unit? In that case, the figure is not so meaningful.

**[Authors] We believe that the AUC score, indeed high, are the results of the imbalance in the landslide impact, with many unaffected buildings and roads. To address this issue we used the F1 score as it takes into account both precision and recall, which are important metrics for imbalanced datasets.**

In the discussion, it is mentioned that the modeling was only done for the 2015 Gorkha earthquake that occurred on 25 April 2015, excluding the event of 12 May 2015. How did you separate the landslides caused by these two earthquakes? I understood that due to the fact that they occurred close to each other, the mapping of the co-seismic landslides could not differentiate well between the landslides caused by both events.

**[Authors] This topic was indeed discussed during the project but, as you mentioned, the two landslide inventories couldn't be differentiated and, therefore, an "aggregated" landslide inventory combining the two events was used by default.**

For the modeling of earthquake-induced landslides it might be considered to apply Spatio-temporal data-driven modeling (e.g. Dahal, A., Tanyas, H., van Westen, C., van der Meijde, M., Mai, P. M., Huser, R. and Lombardo, L. (2024c) Space-time landslide hazard modeling via ensemble neural networks. Natural Hazards and Earth System Sciences 24(3), 823–845. It might be good to address this a bit more in the discussion section.

In how far do the interpolated PGA values correctly represent the effect of topography, and would PGA values be the best predictor for landslide occurrence alone (See also: Dahal, A., Tanyaş, H. and Lombardo, L. (2024b) Full seismic waveform analysis combined with transformer neural networks improves coseismic landslide prediction. Communications Earth & Environment 5(1), 75.)

**[Authors] Thank you for the insightful references. Our current framework is indeed focused on capturing the immediate cascading hazards post-earthquake, particularly their effects on infrastructure. While spatio-temporal data-driven models like those discussed by Dahal et al. (2024c) and Dahal et al. (2024b) represent advanced methods for improving landslide susceptibility predictions, we consider them to be complementary to—but currently beyond the scope of—this initial case study.**

**However, we fully acknowledge that integrating such techniques into our framework could greatly enhance the overall multi-hazard cascade modelling. The inclusion of high-resolution Peak Ground Acceleration (PGA) values that account for topographic effects, along with improved landslide modelling approaches, would allow for a more robust prediction of cascading hazards. These improvements would not only refine the understanding of individual hazard triggers but also improve the interaction between sequential processes in the hazard cascade.**

The reference to Fan et al. 2019 is missing in the reference list.

**[Authors] Amended**